# Anatomical Features of the Deep Cervical Lymphatic System and Intrajugular Lymphatic Vessels in Humans

**DOI:** 10.3390/brainsci10120953

**Published:** 2020-12-09

**Authors:** Kaan Yağmurlu, Jennifer D. Sokolowski, Musa Çırak, Kamran Urgun, Sauson Soldozy, Melike Mut, Mark E. Shaffrey, Petr Tvrdik, M. Yashar S. Kalani

**Affiliations:** 1Department of Neurological Surgery, University of Virginia Health System, Charlottesville, VA 22903, USA; ky7zb@virginia.edu (K.Y.); JDE2Z@hscmail.mcc.virginia.edu (J.D.S.); musacirak@hotmail.com (M.Ç.); drkamranurgun@gmail.com (K.U.); SS2AH@hscmail.mcc.virginia.edu (S.S.); melikem@hacettepe.edu.tr (M.M.); mes8c@hscmail.mcc.virginia.edu (M.E.S.); PT8BM@hscmail.mcc.virginia.edu (P.T.); 2Department of Neuroscience, University of Virginia Health System, Charlottesville, VA 22903, USA; 3Department of Neurosurgery, Hacettepe University, P.O. Box 06230 Ankara, Turkey; 4Department of Neurosurgery, St. John’s Neuroscience Institute, School of Medicine, University of Oklahoma, Tulsa, OK 74104, USA

**Keywords:** deep cervical lymph nodes, meningeal lymphatics, neurolymphatic system, jugular foramen, neck anatomy, lymphatic channels, head and neck cancer, metastasis

## Abstract

Background: Studies in rodents have re-kindled interest in the study of lymphatics in the central nervous system. Animal studies have demonstrated that there is a connection between the subarachnoid space and deep cervical lymph nodes (DCLNs) through dural lymphatic vessels located in the skull base and the parasagittal area. Objective: To describe the connection of the DCLNs and lymphatic tributaries with the intracranial space through the jugular foramen, and to address the anatomical features and variations of the DCLNs and associated lymphatic channels in the neck. Methods: Twelve formalin-fixed human head and neck specimens were studied. Samples from the dura of the wall of the jugular foramen were obtained from two fresh human cadavers during rapid autopsy. The samples were immunostained with podoplanin and CD45 to highlight lymphatic channels and immune cells, respectively. Results: The mean number of nodes for DCLNs was 6.91 ± 0.58 on both sides. The mean node length was 10.1 ± 5.13 mm, the mean width was 7.03 ± 1.9 mm, and the mean thickness was 4 ± 1.04 mm. Immunohistochemical staining from rapid autopsy samples demonstrated that lymphatic vessels pass from the intracranial compartment into the neck through the meninges at the jugular foramen, through tributaries that can be called intrajugular lymphatic vessels. Conclusions: The anatomical features of the DCLNs and their connections with intracranial lymphatic structures through the jugular foramen represent an important possible route for the spread of cancers to and from the central nervous system; therefore, it is essential to have an in-depth understanding of the anatomy of these lymphatic structures and their variations.

## 1. Introduction

Studies in rodents have re-kindled interest in the study of lymphatics in the central nervous system [1,2,3]. Despite convincing evidence in rodent models, the demonstration of lymphatic equivalents in human and their connection with the peripheral lymphatic system has remained elusive. Animal studies have demonstrated that there is a connection between the subarachnoid space and deep cervical lymph nodes (DCLNs) through dural lymphatic vessels located in the skull base and parasagittal areas [2,4,5,6,7,8,9,10]. It is hypothesized that two additional connections exist, one through the cribriform plate and one around the internal jugular vein, which may allow communication between dural lymphatics and the peripheral lymphatic system [2,11,12]. However, the connections between the lymphatic system of the central nervous system (CNS) and DCLNs have not been definitively described in humans. There have been some case reports describing glioblastoma metastasis to the DCLNs with an unclear mechanism and pathway [13]. To the best of our knowledge, a connection between the dural lymphatics and the neck lymphatics has not been previously demonstrated in humans. In this study, we focus on the anatomical features and variations of deep cervical lymph nodes, and their connections with the intracranial lymphatic system through the jugular foramen in humans.

## 2. Methods

This study was conducted in the University of Virginia Skull Base Laboratory. The human cadavers were obtained by the Virginia State Anatomical Program (VSAP Form #2004 and code 1950). Twelve (24 sides) formalin-fixed, silicone-injected adult cadaveric heads without any intracranial, extracranial, or sinonasal pathology were examined [14]. Dissections were performed under an operating microscope (Zeiss OPMI/Neuro NC4, Carl Zeiss^®^, Oberkochen, Germany) to expose the descriptive anatomy in the region of interest. The dissections were photographed using a digital camera (Canon^®^ EOS 5D, Mark IV, Long Island, NY, USA). The mean numbers of lymph nodes on both sides were determined, and a digital ruler was used for the measurements of lymph node length, width, and thickness. Fresh tissues from dura around the jugular foramen and vertebral artery were obtained from two cadavers during rapid autopsy. Tissue was fixed in 4% paraformaldehyde for 48 hours and then paraffin processed via standard techniques. Sections were cut at 10 µm, dewaxed, and rehydrated through sequential incubation in xylenes and graded alcohol baths. Antigen retrieval was performed by heating buffer (10 mM Tris, 1 mM EDTA, pH 9) to 80 °C, adding slides, and allowing them to cool to room temperature. The sections were blocked (in 2% normal goat serum and 0.1% tween in PBS) for 1 h at room temperature and then incubated in primary antibodies diluted in blocking buffer, overnight at 4 °C. Lymphatic endothelial cells were labelled using an anti-podoplanin antibody (PDPN, mouse monoclonal, D2-40, 1:100, BioLegend^®^, San Diego, CA, USA), and immune cells were labelled via an anti-CD45 antibody (rabbit polyclonal, ab10558, 1:150, Abcam^®^, Cambridge, MA, USA). Alexa-fluor-conjugated secondary antibodies were used for visualization (1:1000, ThermoFisher^®^, Middletown, VA, USA). The slides were coverslipped with DAPI-containing mounting medium (Prolong Gold Antifade, ThermoFisher^®^, Middletown, VA, USA) to highlight nuclei. Consent was obtained from the institutional review board (IRB) (Code 1950) for the harvest of fresh tissue from cadavers during rapid autopsy.

## 3. Results

### 3.1. Lymph Node Location

The deep cervical lymph nodes are located around the internal jugular vein. Based on the classification of the neck lymph nodes, the DCLNs include groups II (jugulodigastric), III (mid-jugular), and IV (low jugular) and are bordered posteriorly by the posterior edge of the sternocleidomastoid muscle, anteriorly by the posterior edge of the submandibular gland, superiorly by the skull base (jugular fossa), inferiorly by the clavicle, and medially by the internal or common carotid artery (Figure 1) [15]. Group II is further subdivided into II A, which is located behind the internal jugular vein, and II B, which is situated in front of the internal jugular vein. The border between groups II and III is the axial level of the lower margin of the hyoid bone. The border between groups III and IV is the axial level of the lower margin of the cricoid cartilage arch. The lymph nodes in the neck, except the DCLNs, are found by following the branches of the external carotid artery, such as the occipital and facial arteries. The mean number of DCLNs in the 24 cadaveric specimens was 6.91 ± 0.58 on both sides. The mean nodal length was 10.1 ± 5.13 mm, the mean width was 7.03 ± 1.9 mm, and the mean thickness was 4 ± 1.04 mm.

### 3.2. Lymphatic Pathway

The jugular foramen is located between the temporal and occipital bones [16]. It has venous and neuronal parts. The venous part has a sigmoid and a petrosal portion. The two parts are divided by the intrajugular process. Cranial nerves IX, X, and XI pass in the neuronal part located between the sigmoid and petrosal portions.

The lymphatic vessels interconnecting the DCLNs are located around and along the internal jugular vein and within the carotid sheath (Figure 2). Some lymphatic vessels arising from the uppermost DCLN turn backward at the level of the jugular foramen to reach the retropharyngeal lymph nodes, while some continue upward into the jugular foramen. Although it was difficult to follow these lymphatic channels through the jugular foramen under the operating microscope, we visualized these lymphatic vessels, which can be called intrajugular lymphatic vessels, with immunostaining techniques (Figure 3). To the best of our knowledge, this is the first demonstration of the lymphatic vessels within the dura of the jugular foramen and evidence of a point of connection between dural lymphatics and the deep cervical lymph nodes in humans.

## 4. Discussion

### 4.1. Deep Cervical Lymph Node Anatomy, Variations, and Clinical Importance

#### 4.1.1. Lymphatics and the Nervous System

We studied the morphological features of the deep cervical lymph nodes and the lymphatic channels that interconnect between them. The lymphatic system contains the capillary network (mean range caliber, 0.01–0.2 mm), pre-collecting and collecting lymph vessels or channels (mean range calibers, 0.2 mm and 0.1–2 mm, respectively), lymphatic trunks (mean caliber, 1.5–3 mm) and ducts (mean caliber, 2–4 mm), and lymph nodes [17,18,19]. Lymphatic networks are located in superficial layers, such as the skin, mucosa, etc. Lymphatic valves are present along the lengths of the pre-collecting and collecting vessels, lymphatic trunks, and lymphatic ducts, which make the vessels look like a string of beads. Another study noted rich avalvular and well-organized lymphatic channels in the nasal fossa [20]. In our study, the lymphatic channels between the DCLNs, which correspond to collecting vessels, had a valve system, seen as a string of beads. The collecting vessels enter and leave lymph nodes. The lymph nodes have various numbers of afferent and efferent collecting vessels. The lymphatic flow is in an up-to-down direction, and the valves are oriented to open downward. In the end, the lymphatic duct drains to the venous angle located in the junction of the subclavian and internal jugular veins on both sides.

#### 4.1.2. Lymph Node Yield

An analysis of the literature encompassing 5228 neck dissections revealed a mean yield of 35 cervical lymph nodes in radical neck dissection specimens [21]. A cadaveric study quantified cervical lymph nodes based on nodal levels [22]. In a sample of 28 cadavers, yielding 56 specimens of modified radical neck dissections, a mean number of 44.4 (Standart Deviation-SD = 16) total lymph nodes were identified. With respect to DCLNs, the mean numbers when averaging both sides of the neck were found to be 7.2 (6.0–8.5, SD = 4.6), 6.5 (5.5–7.4, SD = 3.6), 6.6 (5.7–7.4, SD = 3.2), and 8.6 for levels IIA, IIB, III, and IV, respectively. Mann–Whitney tests identified a significant difference only in group IV nodes (*P* = 0.026). In addition, the mean number of group IV nodes was found to be higher in males (11.9) than in females (7.2) (*P* = 0.040), thus guiding the surgeon in their neck dissection with respect to the patient demographics at hand. In the current study, regardless of the gender, the mean nodal number for DCLNs was 6.91 ± 0.58 on both sides, a number that is relatively smaller than that noted in the literature. Additionally, we measured the lengths, widths, and weights of the DCLNs (Table 1).

#### 4.1.3. Lymph Node Biopsy and Dissection

An atlas to guide lymph node dissection and biopsy is an important tool for head-and-neck and general surgeons. One study highlighted the variability in nodal yield with respect to differences in radical neck dissection techniques among surgeons, making it difficult to provide consistent pathology specimens for analysis, as well as to correlate patient outcomes with nodal yields [21]. They identified a mean nodal yield of 32 (SD = 15.3) from 64 cadaveric radical neck dissections [21]. The differences in and underestimation of nodal yield can be attributed to a variety of factors, including those related to patient characteristics, surgical techniques, or pathologic examination.

#### 4.1.4. Metastases to Deep Cervical Lymph Nodes

The DCLNs are often the site of metastases from head and neck malignancies, and the presence of metastases is a critical prognostic factor in predicting patient survival. In squamous cell carcinoma (SCC) of the head and neck, the extent of cervical lymph node metastases has been shown to be the most important factor in determining patient outcomes [23]. Metastatic assessment is performed based on lymph node biopsies sent for pathologic evaluation, the quality of which is based on lymph node yield. The TNM staging system developed by the Union for International Cancer Control and the “Cancer staging manual” by the American Joint Committee on Cancer recommend at least 10 nodes for radical neck dissection, and at least six for selective neck dissection for adequate histopathological examination [24]. For this reason, an adequate understanding of not only anatomic distribution but also lymph node quantification is important when retrieving lymph nodes for pathological assessment. The results of these pathology reports help to guide treatment plans, providing indications for surgery, adjuvant therapy, or standalone radiation or chemotherapy. Different surgical approaches, especially “berry picking” by inexperienced surgeons, can influence yield in a negative manner [25]. This highlights the need for a standard approach and baseline count of nodal yield, necessitating further cadaveric studies that quantify DCLNs.

Clinically, intradural lymphatic vessels may represent a pathway for the spread of metastases. The presence of cervical lymph node metastases from glioblastoma has been shown [13]. Metastasis in glioblastoma is speculated to occur in patients with repeated craniotomies, which may allow tumor access to superficial lymphatic structures, which would then drain to deep cervical lymph nodes; however, this does not explain the extraneural metastases that occur prior to surgery [13]. On the other hand, some malignant pituitary carcinoma metastases to the deep cervical lymph nodes have been reported and postulated to occur as a result of the local spreading of the tumor to reach the superficial lymphatic network and then deep cervical lymph nodes [26]. Instead, it is possible that these metastases occur via the intrajugular lymphatic vessels through the jugular foramen. We speculate that the intrajugular lymphatic vessels may be a possible route for the metastasis of the intracranial tumors to the deep cervical lymph nodes.

### 4.2. Deep Cervical Lymph Nodes and Connections to the Intracranial Compartment through the Jugular Foramen

#### Routes of Lymphatic Drainage

In non-human mammals, the pathways for the cerebrospinal fluid (CSF) outflow include the nasal mucosa, the cribriform plate, and the arachnoid villi. Other possible routes of lymphatic drainage are the Virchow–Robin spaces around the arteries, veins, and the cranial and spinal nerves, and dural lymphatics around the jugular foramen, which has been outlined in rodents [27,28,29]. The daily amount of CSF production in animals is significantly less than in humans. Although most of the studies regarding CSF and interstitial fluid drainage pathways come from animal experiments, there are major differences between animals and humans in the pathways of CSF drainage. In most animals, up to 50% of CSF drains to cervical lymph nodes, whereas a significant proportion of CSF egress in adult humans likely drains directly into venous blood via arachnoid villi and granulations [27,30,31]. Overall, animal data indicate that lymphatic vessels are present in the dura mater of the CNS and drain out of the skull via the foramina of the base of the skull alongside arteries, veins, and cranial nerves [2]. To date, there has been no unequivocal proof of CSF drainage through the cribriform plate to the nasal mucosa or through the jugular foramen to the neck cervical lymph system in humans. We think that a minor amount of CSF may drain into the DCLNs by means of the intrajugular lymphatic vessels in humans. It has been proposed that the meningeal lymphatic vessels drain CSF and interstitial fluid to the cervical lymph nodes [2,5,6,7,8,9,10]. In rodent studies, the meningeal lymphatic vessels in the skull base (basal lymphatic vessels) and between the dural layers along the superior sagittal sinus (dorsal lymphatic vessels) are considered the main route for CSF drainage, especially the clearance of macromolecules, to the cervical lymph nodes. The meningeal lymphatic vessels have direct connections with extracranial lymphatic vessels through the skull foramina [4]. One group, using a T2-FLAIR magnetic resonance sequence, has demonstrated CSF egress from the ventricles to the deep cervical lymph nodes through the jugular foramen [32]. There are two different opinions about the lymphatic outflow through the skull base: The first one is that the lymphatic vessels follow the perineural routes through foramina [33]. The second suggests that the lymphatic vessels are located apart from the nerve fibers, and the lymphatic vessels are still intact around the foramina, especially in the jugular foramen, even after the removal of the cranial nerves, which proposes that the lymphatic vessels have distinct anatomical locations relative to the perineural lymphatic system [4]. We found the intrajugular lymphatic vessels, the lymphatic vessels located within the dura around the jugular foramen, but not around the cranial nerves passing through the foramen. We did not observe any lymphatic structure around the venous part of the foramen. The lymphatic vessels from the intracranial space run within the dura around the jugular foramen rather than within the foramen, and then connect with the deep cervical lymphatic network around the internal jugular vein.

This is the first anatomical description and demonstration that there are connections between neck lymphatic structures and intracranial lymphatic structures by means of intrajugular lymphatic vessels that run through the jugular foramen. The lymphatic vessels from the intracranial space run through the dura rather than immediately adjacent to the venous or neural structures, and these intrajugular lymphatic vessels appear to connect with the deep cervical lymphatic network around the internal jugular vein. This finding lends further support to the notion that lymphatic structures drain the intracranial compartment of humans, as has been demonstrated in rodents.

## 5. Conclusions

Understanding the morphology and anatomy of the lymphatic connection between deep cervical lymph nodes and the intracranial space is crucial for identifying the etiopathogenesis of neurodegenerative diseases and cancer metastasis to and from the brain. In the case of deep cervical lymph node metastasis, an adequate understanding of not only anatomic distribution but also lymph node quantification is important when retrieving lymph nodes for pathological assessment.

## Figures and Tables

**Figure 1 brainsci-10-00953-f001:**
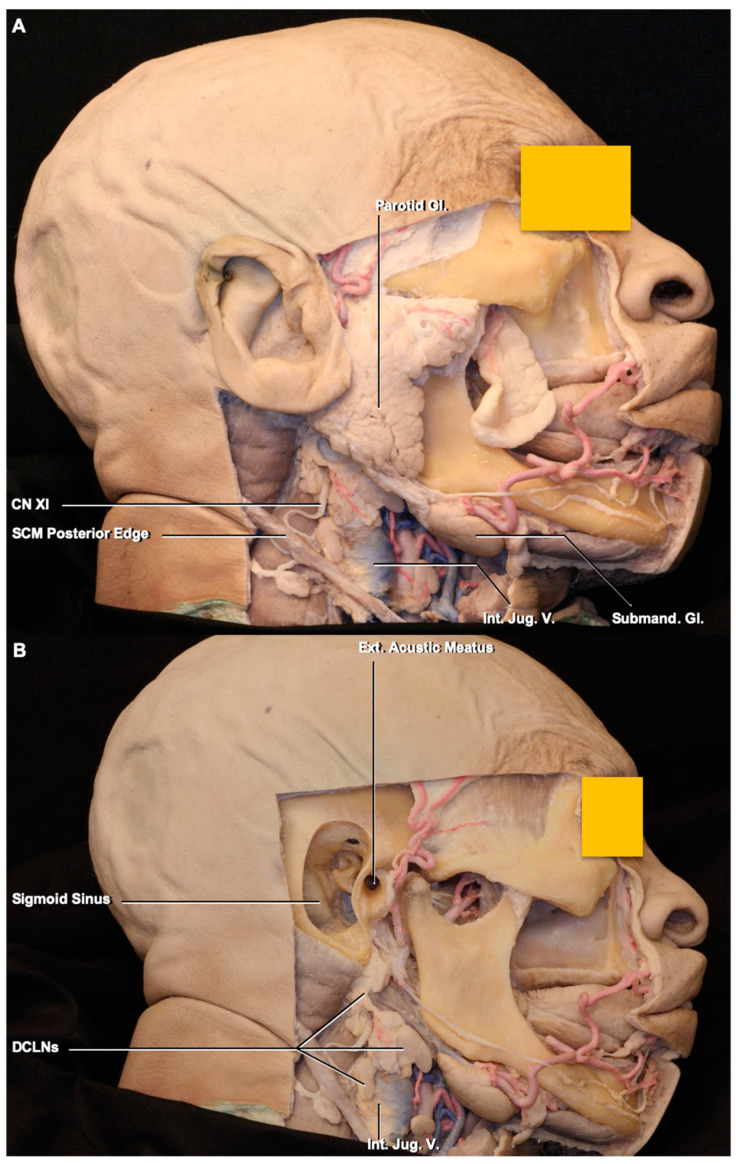
Stepwise dissections of the face and neck. (**A**), parotid gland was exposed after removal of facial skin and subcutaneous fat tissue. In the neck, the sternocleidomastoid muscle except its posterior edge, and platysma have been removed to expose the internal jugular vein and deep cervical lymph nodes. (**B**), after removal the external ear, mastoid bone, and parotid gland. The deep cervical lymph nodes are exposed. (**C**), groups of the deep cervical lymph nodes. The deep cervical lymph nodes are located in groups II, III, and IV in the neck. Abbreviations: CN, cranial nerves; DCLNs., deep cervical lymph nodes; Ext., external; Gl., gland; Int., internal; Jug., jugular; LN., lymph node; SCM, sternocleidomastoid muscle; Submand., submandibular; V., vein.

**Figure 2 brainsci-10-00953-f002:**
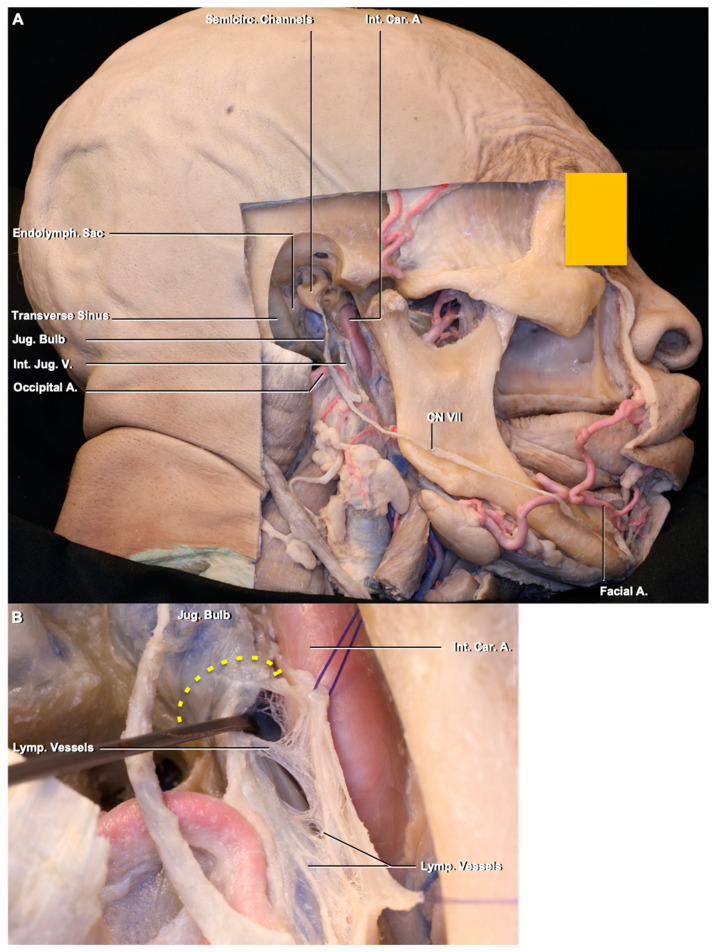
(**A**) Middle ear: the jugular and carotid foramina were drilled to expose the jugular fossa, uppermost part of the internal jugular vein, and jugular bulb. The lymphatic vessels connecting the lymph nodes located around the internal jugular vein within the carotid sheath. (**B**) Enlarged view of the jugular fossa and lymphatic vessels exposed in the jugular fossa. The majority of the lymphatic vessels turn back to reach the retropharyngeal lymph nodes. (**C**,**D**) Enlarged view of the shaded area; the collecting lymphatic vessels were exposed in a shape of string of beads formed by the lymphatic valves. Abbreviations: A., artery; Car., carotid; CN, cranial nerve; Endolymp., endolymphatic; Int., internal; Jug., jugular; Lymp., lymphatic; LN, lymph node; Semicirc., semicircular; V., vein.

**Figure 3 brainsci-10-00953-f003:**
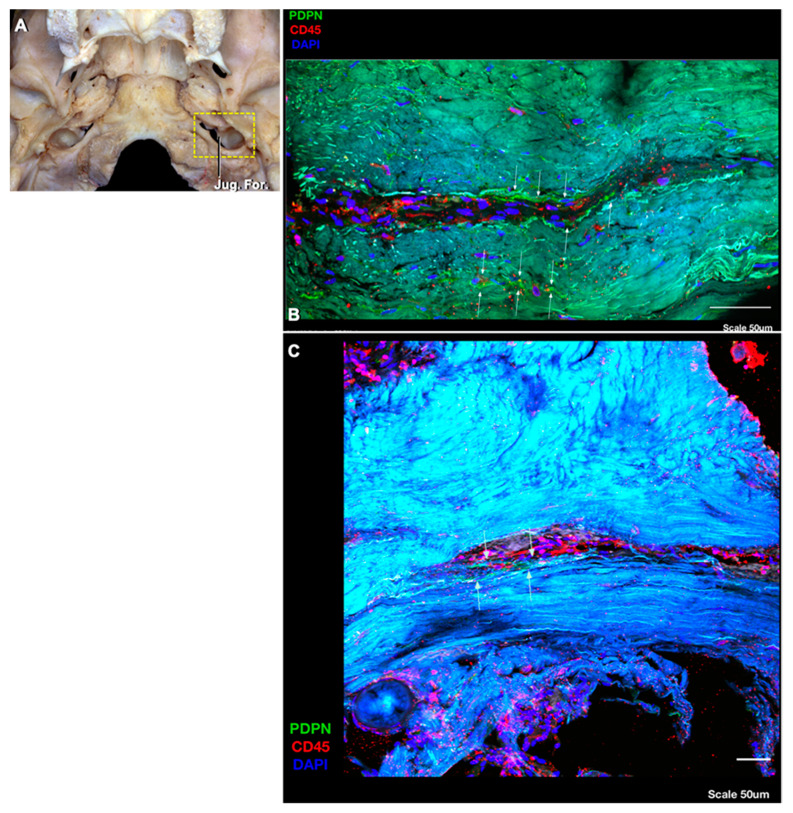
Immunohistochemical staining of the wall of the jugular foramen. (**A**) An inferior view of the skull. The shaded area indicates the jugular foramen and location of sample harvest. (**B**) The immunohistochemical staining of the wall of the jugular foramen, the lymphatic structure (white arrows) along the wall stained positively with PDPN antibody (green colored), and immune cells (T cells) stained by CD45 antibody (red colored) within the lymphatic structure. The DAPI marks the nuclei. (**C**) Another specimen; the lymphatic vessels (white arrows) were seen with staining of PDPN and CD45. Abbreviations: DAPI, 4′6-diamidino-2-phenylindole; For., foramen; Jug., jugular; PDPN, podoplanin.

**Table 1 brainsci-10-00953-t001:** Size of the deep cervical lymph nodes.

DCLN Groups	Lenght (mm) Mean, SD, raNge	Width (mm) Mean, SD, Range	Weight (mm) Mean, SD, Range
IIA	17.09 ± 4.20 (14.42–22.05)	6.50 ± 1.38 (4.47–8.21)	4.25 ± 0.86 (3.00–5.29)
IIB	7.99 ± 5.33 (3.12–17.27)	6.30 ± 1.98 (2.71–9.01)	3.28 ± 1.03 (2.19–4.48)
III	12.30 ± 5.42 (4.11–20.41)	8.72 ± 2.43 (5.41–13.79)	4.48 ± 2.06 (0.91–7.91)
IV	9.65 ± 1.54 (8.08–12.21)	4.50 ± 1.28 (3.00–6.54)	3.37 ± 1.20 (1.71–5.00)

SD: Standart Deviation.

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
