# Peer review of "Anatomical Features of the Deep Cervical Lymphatic System and Intrajugular Lymphatic Vessels in Humans"

_brainsci, 2020, doi:10.3390/brainsci10120953_

Round 1

Reviewer 1 Report

The study "Anatomical Features of the Deep Cervical Lymphatic System and Intrajugular Lymphatic Vessels in Human" performed by Yağmurlu et al., is an interesting story.

  1. However authors have not provided enough evidence in the form of results to support their claims.
  2.  The discussion section is also very diffuesd in its present form and need to be rewritten. 

Author Response

Reviewer 1.

The study "Anatomical Features of the Deep Cervical Lymphatic System and Intrajugular Lymphatic Vessels in Human" performed by Yağmurlu et al., is an interesting story.

  1. However authors have not provided enough evidence in the form of results to support their claims.

  1.  The discussion section is also very diffuse in its present form and need to be rewritten. 

#Response: We thank the review for their comments. This is the first comprehensive anatomic dissection of the deep cervical lymphatics. We provide detailed data on the number, size and relative location of nodes. This information is novel and unique. Detailed dissections of the lymphatic are rare in the literature. Although we have made minor revisions in the manuscript.

Reviewer 2 Report

In this manuscript the authors demonstrate for the first time the lymphatic vessels within the dura of the jugular foramen and provide evidence of a connection between dural lymphatics and the deep cervical lymph nodes (DCLNs) in humans. This is a highly relevant and useful study and will be helpful especially as DCLNs are frequent sites of metastases.

The introduction and discussion have been well written. I have following suggestions:

  1. The manuscript needs to be thoroughly rechecked for English and syntax errors. In several places DCLN has been written as DLCN. On line 203 it should be animals. Should the word ‘man’ be replaced with human in line 37? Also check line 225 and 226.
  2. Where is Table 1? (line 159)
  3. The immunohistochemical staining is not clearly visible. Please provide higher magnification images. Under the results section the observations for Figure 3 needs to be described.
  4. Because of the green autofluorescence it is hard to see PDPN staining.
  5. The authors need to show the ‘string of beads’ arrangement clearly in the figures.
  6. Do authors plan to perform any study using dyes to establish their theory?

Author Response

Reviewer 2.

In this manuscript the authors demonstrate for the first time the lymphatic vessels within the dura of the jugular foramen and provide evidence of a connection between dural lymphatics and the deep cervical lymph nodes (DCLNs) in humans. This is a highly relevant and useful study and will be helpful especially as DCLNs are frequent sites of metastases.

#Response: We really appreciate the reviewer for a meticulous and detailed revision of our paper.

The introduction and discussion have been well written. I have following suggestions:

  1. The manuscript needs to be thoroughly rechecked for English and syntax errors. In several places DCLN has been written as DLCN. On line 203 it should be animals. Should the word ‘man’ be replaced with human in line 37? Also check line 225 and 226.

#Response: We appreciate the reviewer for warning about these syntax errors. We have corrected all “DLCN” as “DCLN”. We have corrected the mistake in line 203. We replaced the man with human in line 37.

  1. Where is Table 1? (line 159)

#Response: Table 1 has been cited in the line 165, but have included it in line 159 as well.